# Rate Control Technology for Next Generation Video Coding Overview and Future Perspective

Hao Zeng [1], Jun Xu [1], Shuqian He [1,*], Zhengjie Deng [1,*] and Chun Shi [2]

1 School of Information Science and Technology, Hainan Normal University, Haikou 571199, China
2 School of Electronic and Information, Guangdong Polytechnic Normal University, Guangzhou 510665, China
* Correspondence: hsq@hainnu.edu.cn (S.H.); hsdengzj@163.com (Z.D.)

**Abstract:** Video data have become the main data traffic on the Internet, and their traffic is increasing explosively every year, thus increasing the pressure of video transmission. Video coding technology has become the key to compressing original videos. As an indispensable technology, rate control plays an important role in stabilizing video stream transmission. Rate control (RC) is part of rate distortion optimization (RDO) whose job is to find the optimal solution based on balancing rate and distortion. It not only needs to consider the buffer and network status but also adjust the corresponding bit rate according to the video content. This paper reviews the related technologies of rate control under high efficiency video coding (HEVC) and versatile video coding (VVC) standards so that subsequent researchers can quickly understand the field and promote the development of rate control algorithms. Firstly, the paper summarizes the various aspects of RC, including basic principles, rate-distortion models, major processes, and performance criteria. Secondly, the paper surveys, in detail, the research progress in the field of rate control and analyzes several mainstream research directions. Thirdly, we carry out relevant experiments on the standard reference software and analyze and discuss the experimental results of the existing studies. Finally, we look ahead to the future trends of rate control and provide feasible improvement suggestions.

**Keywords:** video coding; rate control; bit allocation; high efficiency video coding; versatile video coding





## 1. Introduction

Under the continuous influence of the novel coronavirus pneumonia epidemic, people are gradually staying at home for longer periods of time. In order to build a bridge of communication between people, the volume of video-related services such as short video and video conferencing has been gradually increasing. According to reports [1–3], mobile web traffic has doubled in the past two years since the first quarter of 2020, and 80% of all videos created in 2021 are user-generated; in 2022, video resources are estimated to account for 82% of all IP traffic. Uncompressed video is huge and usually needs to be encoded for transmission. In recent decades, in order to achieve high-quality, high-resolution video storage and transmission, the Video Coding Experts Group (VCEG) of the International Telecommunication Union Telecommunication Standardization Sector (ITU-T) and the Moving Picture Experts Group (MPEG) of the International Organization for Standardization (ISO)/International Electrotechnical Commission (IEC) has released a series of international video coding standards: from the previous Advanced Video Coding (AVC/H.264) [4] and HEVC/H.265 [5] standards to the latest video coding standard (VVC/H.266) [6]. Although each standard has significant changes, a hybrid coding framework consisting of modules such as block segmentation, intra-frame prediction, inter-frame prediction, transform, quantization, entropy coding, and loop filtering is still used. The encoder will then use this hybrid coding framework to remove the temporal, spatial, and information entropy redundancy in the unprocessed video to achieve high-performance video compression. The main changes of the new generation standard are as follows: The HEVC standard introduces a quad-tree-based block division scheme, supports 35 intra-frame prediction modes

(including 33 types of angle prediction), provides new inter-frame prediction technologies (including motion information fusion technology (Merge), motion vector prediction technology (AMVP) and merge-based skip mode), and realizes the compensation of reconstructed pixel values through sample adaptive offset (SAO) technology and the use of advanced context-based adaptive binary arithmetic coding (CABAC) for entropy coding, etc. The VVC standard adds binary and ternary tree schemes to block division and supports 67 intra-frame prediction modes (including 65 angle prediction modes). Inter-frame prediction also adds an extended Merge mode and bidirectional optical flow technology. Affine motion compensated prediction and other technologies add luma mapping with chroma scaling (LMCS) and an adaptive loop filter (ALF) to the loop filter. In the hybrid video coding framework, rate control is not added as a separate module, but this does not mean that rate control is not important.

Rate control, as the name implies, controls the number of bits transmitted during encoding. Video transmission will be affected by bandwidth and network fluctuations, so a reasonable allocation of bit rates can improve the quality of transmission. In many video scenarios, such as online on-demand, online live broadcast, and real-time communication, rate control plays an important role. Taking online video and live streaming as an example, it is necessary to limit the bit rate due to the constant bandwidth of the user and the limited amount of data that can be buffered. Otherwise, some clips with an excessively high instantaneous bit rate will cause stuttering and bring a poor experience to the user. A great deal of large-scale live video traffic is distributed to users through a content delivery network (CDN), which pays by traffic, where an unreasonable bit rate allocation will make bandwidth costs uncontrollable. Therefore, the streaming media server often transcodes the video and converts the video source into smooth, standard definition, high definition, ultra-clear, and other grades of video through rate control technology for people to watch conveniently. In general, for each different business scenario, the various restrictions on the bit rate will also be different. The goal of rate control is to achieve scene optimization by controlling each frame or basic coding unit under the premise of ensuring coding efficiency and subjective quality.

In this paper, we aim to provide a comprehensive overview of rate control techniques from HEVC to VVC. The main contributions of this work are as follows: (1) The detailed structure of rate control from HEVC to VVC is organized, from the basic principles to the performance indicators, so that readers can understand the whole picture of rate control. (2) The research status of rate control in recent years is summarized, including the optimal bit allocation for rate-distortion models, perceptual rate control, heuristic rate control, rate control under artificial intelligence, and rate control under extended standards. At the same time, related algorithms are also introduced and their advantages and disadvantages are discussed. (3) Finally, suggestions are made for future development exploration.

The rest of this paper is organized as follows: Section 2 presents an overview of rate control from multiple aspects, Section 3 describes the current state of rate control research from different perspectives, comparative experiments are discussed in Section 4, and finally, Sections 5 and 6 summarize the paper and make constructive suggestions.

## 2. Rate Control

### 2.1. Basic Principle

Rate control selects the coding parameters to make the output rate of the encoder equal to the pre-set target rate while minimizing the coding distortion and can be expressed as an optimization problem under strict rate constraints:

$$\underset{\text{Para}}{\arg\min} D \text{ s.t. } R \leq R_{\text{tar}} \tag{1}$$

where Para represents the encoding parameter set, D is the encoding distortion, and R and $R_{\text{tar}}$ are the encoder output bit rate and target bit rate, respectively. It is not difficult to see from the above equation that rate control is inseparable from rate-distortion optimization.

Rate-distortion theory can be traced back to Shannon's "Discrete Source Coding Theorem under Fidelity Criterion" [7]. Based on this theory, a rate-distortion model is established, the target bit rate is determined, and then the actual bit rate is allocated according to the rate-distortion model, which is the process of rate control. The overall goal of rate control is to control the number of bits of each frame or basic coding unit so as to minimize the distortion under the condition of a certain total number of bits so each buffer will not overflow (the details are shown in Figure 1). As each frame of the video sequence passes through the encoder, the encoder analyzes the content and buffer state and sequentially performs bit allocation and bit implementation to guide the encoding, allowing the generated media stream to enter the channel.

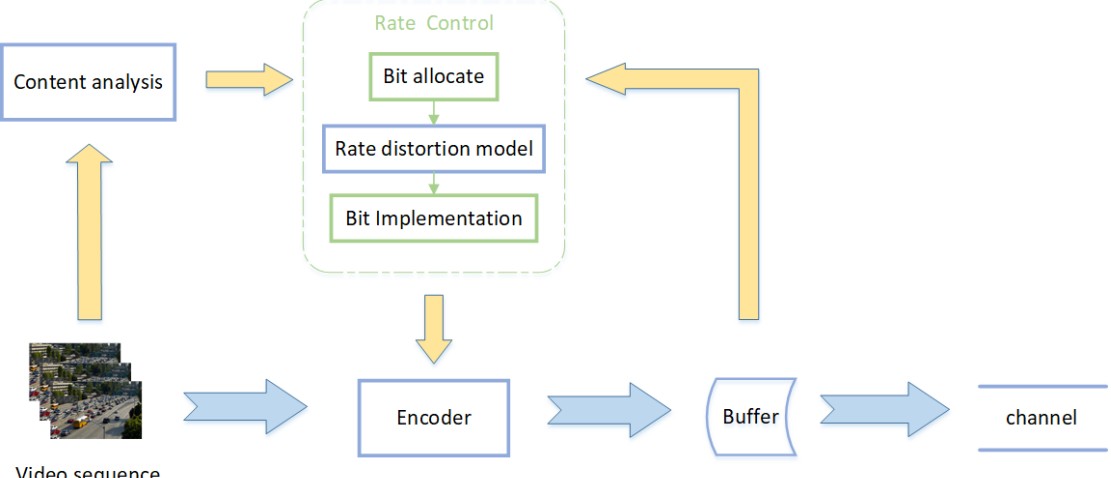

**Figure 1.** The overall framework for rate control.

*2.2. Rate-Distortion Model*

2.2.1. R-Q Model

The rate-distortion model of rate control refers to the R-Q model in a broad sense, where R stands for bit rate and Q stands for quantization parameter (QP). However, in order to distinguish the direct or indirect connection between R and Q, in a narrow sense, we use the model with a direct connection between R and Q, the R-Q model. Rate control algorithms based on the R-Q model have been widely studied in the past. The TMN8 algorithm of H.263 [8], VM5 algorithm of MPEG-4 [9], JVT-G012 proposal [10] algorithm of H.264, and JCTVC-H0213 proposal [11] algorithm of H.265 all suggested the relevant R-Q model, and the early reference software version of HEVC(HM6.0) also demonstrated the feasibility of the Unified R-Q (URQ) model [12]. The R-Q model can be mainly divided into the linear R-Q model and the quadratic R-Q model. The relationship of the linear R-Q model is as follows:

$$R = \frac{\alpha * S}{Q} \tag{2}$$

$$p(x) = \frac{\alpha}{2} e^{-\alpha|x|} \tag{3}$$

$$R = MAD \times \left( \frac{\alpha}{Q_{step}} + \frac{\beta}{Q_{step}^2} \right) \tag{4}$$

where $\alpha$ is the parameter of the linear regression and S is the complexity of the target source. The quadratic R-Q model is widely used in the R-Q model. The theory of the R-Q model comes from the classic rate-distortion function [13]. The discrete cosine transform (DCT) coefficient obtained by the early standard obeys the Laplace distribution as shown in Equation (3). The R-Q relationship in the quadratic R-Q model is expressed as shown in Equation (4). $Q_{step}$ represents the basic quantization step size, $\alpha$ and $\beta$ are coding

parameters, which will be updated with a coding unit is encoded, and the mean absolute deviation (MAD) is the absolute error of the predicted coding unit, which can measure the complexity of the coding unit through residual information.

In MPEG.4 rate control, α, β, MAD, and the target number of bits can be obtained by performing a quadratic R-D model before estimating QP. Different from the MPEG series, it is more difficult to rate control the H.26X series by this method than the previous method because the quantization parameters are not only used in the rate control algorithm but also applied in the rate-distortion optimization. This creates "the chicken and egg dilemma" [14], meaning that RC conflicts with the RDO process. In order to solve this dilemma, a linear prediction model is introduced in H.264 to predict MAD, and the equation is as follows:

$$\mathrm{MAD}_{\mathrm{cur}} = a_1 \times \mathrm{MAD}_{\mathrm{pre}} + a_2 \tag{5}$$

where $\mathrm{MAD}_{\mathrm{cur}}$ and $\mathrm{MAD}_{\mathrm{pre}}$ represent the MAD at the corresponding position of the current basic unit and the previous frame, and $a_1$ and $a_2$ are model coefficients with initial values of 1 and 0 and are updated by least squares regression when the last macroblock of each basic unit is processed.

### 2.2.2. R-ρ Model

The rate control algorithm in the ρ-domain was first proposed by He et al. [15]. The algorithm mainly focuses on the relationship between the bit rate (R) and the percentage of zero coefficients (ρ) after quantization. He et al. [15] found that there is a linear relationship between R and ρ. The specific R-ρ relationship is as follows:

$$\mathrm{R} = \theta \cdot (1 - \rho) \tag{6}$$

Wang et al. [16] also proposed the method of quadratic ρ domain on this basis. Although the ρ-domain rate control algorithm can provide a smoother output bit rate and better target quality, it has been gradually abandoned because it does not apply the new standard; it is only applicable to the earlier H.263 standard.

### 2.2.3. R-λ Model

The rate control algorithm based on the R-λ model is the algorithm adopted by the current VVC standard reference software and is also the mainstream algorithm currently studied. The R-λ model accurately refers to the R-λ-QP model. The R-λ-QP model is a model established by Li Bin et al. [17] with λ as the link. This proposal was tried in the HEVC reference software HM8.0 version, achieved good coding performance, and was subsequently formally applied in the HM10.0 version. λ is the slope (absolute value) of the rate-distortion curve and can be calculated by the following equation:

$$\mathrm{D(R)} = \mathrm{C(R)}^{-k} \tag{7}$$

$$\lambda = -\frac{\partial \mathrm{D}}{\partial \mathrm{R}} = \mathrm{CK} \cdot \mathrm{R}^{-K-1} \triangleq \alpha \mathrm{R}^{\beta} \tag{8}$$

The rate-distortion relationship of R-λ model is derived from hyperbolic curve fitting; C and K are the parameters of the hyperbolic model and α and β are parameters related to the content of the sequence. It can be seen from Equation (8) that the bit rate R in the model will be determined by the Lagrange multiplier λ. Since the RDO process is too cumbersome, in order to simplify the QP decision, the rate control algorithm is simplified by the following equation [18]:

$$\mathrm{QP} = 4.2005 \times \ln(\lambda) + 13.7122 \tag{9}$$

The R-λ model has been adopted for a long time, so this paper collates the studies on the proposals related to rate control in the reference software, as shown in Table 1.

**Table 1.** The R-λ model development proposal.

| Coding Standard | Proposal | Main Role |
|---|---|---|
| HEVC | JCTVC-K0103 [19] | First proposed the R-λ model and included it in HM10.0 as the main standard |
| HEVC | JCTVC-M0036 [20] | Adjust the updated parameter range and proposed an adaptive bit allocation scheme |
| HEVC | JCTVC-M0257 [21] | Mainly proposed improvements for intra-frame bit allocation and bit implementation |
| HEVC | JCTVC-U0132 [22] | Avoided buffer overflow and underflow |
| HEVC | JCTVC-V0078 [23] | Adjusted the lower limit of buffer overflow in the U0132 proposal |
| VVC | JVET-K0390 [24] | The skip area and sub-skip area were divided at the frame level for bit allocation |
| VVC | JVET-M0060 [25] | First proposed the quality dependent factor (QDF) for bit allocation |
| VVC | JVET-T0062 [26] | Adopted the QDF and made RC support GOP=32 in the RA configuration |
| VVC | JVET-Y0105 [27] | Updated the skip and non-skip area bit allocation CTU level bit allocation |

### 2.3. The Process of Rate Control

The bit rate control can be divided into two parts, the target bit allocation and the target bit implementation. The target bit allocation assigns a target bit rate to each basic coding unit according to the existing video content and buffer state. Since the actual bit is not equal to the target bit, it is necessary to determine the appropriate QP through the rate control model when the target bit is realized, and then achieve the target bit rate after the quantization parameter, QP, is encoded. The target bit allocation is mainly divided into three levels: the GOP level, frame level, and coding tree unit (CTU) level. The target bit implementation calculates the λ and QP of the frame level and CTU; the specific process is shown in Figure 2.

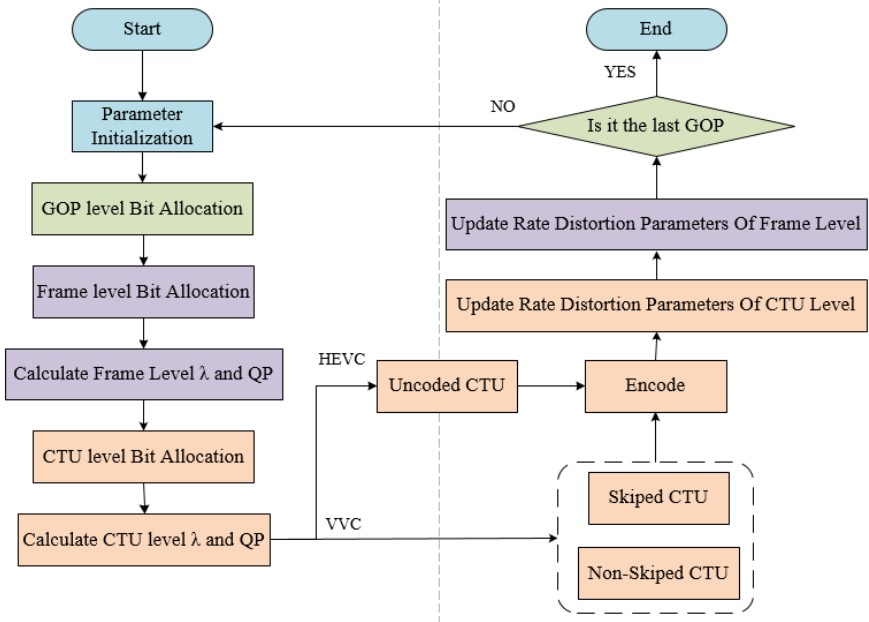

**Figure 2.** The basic flow of rate control based on the R-λ model.

Taking the reference software of VVC as an example, the actual configuration of VTM video coding [28] is usually as follows: all intra (AI), low delay (LD), and random access (RA). In the AI configuration, each image in the video stream is encoded as an IDR frame, which is suitable for scenarios with poor channel environments and prone to packet loss. The LD configuration is suitable for scenarios with high latency requirements, such as video calls and video conferences, and can be divided into low delay B (LDB) and low delay P (LDP) configurations, corresponding to the two previous coding structures: IBBB and IPPP. The characteristics of the two structures are that the video frames are all B frames

or P frames, except that the first frame is an I frame, but here the B frame refers specifically to the generalized P and B (GPB) [29]. The RA coding structure adopts the hierarchical-B structure [30], which is suitable for broadcasting and streaming media applications. Since the AI configuration does not have B and P frames, there is no need to divide the GOP, so it does not need to be specified in the encoding configuration. There are three modes of bit allocation which are equal-bit allocation, fixed-rate bit allocation, and adaptive bit allocation; the latter two are also hierarchical-B structures. CTU level rate control is not necessary; as seen in the study, it can be enabled or disabled in the coding configuration according to specific needs.

*2.4. The Performance Index of Rate Control*

The performance index of the algorithm reflected in the rate control experiment is mainly as follows:

- Rate-distortion performance: The peak signal-to-noise ratio (PSNR) value is often used to represent the distortion introduced by the encoder during the encoding process, but it cannot be directly compared because the actual bit rates are often different. The BD-PSNR (Bjøntegaard-delta-psnr) and BD-rate (Bjøntegaard-delta-rate) are commonly used rate-distortion performance standards [31]. The BD-rate represents the rate increase in the optimized algorithm compared with the original algorithm under the same objective video quality. If the BD-rate is negative, it indicates that the coding performance of the optimized algorithm is improved. Of course, some papers use SSIM instead of PSNR to calculate BD-SSIM.
- Bit allocation error (BRE)/Bit allocation accuracy (BRA): From the previous section, we know that QP is an integer, and the actual bit rate and the target bit rate are always different. An important index of the rate control algorithm is the precision of rate control; the higher the precision of rate control, the better the algorithm. In other words, the smaller the error of rate allocation, the better the algorithm. The bit allocation error/bit allocation accuracy is calculated from the target bit rate and the actual bit rate as follows:

$$\text{BRE} = \frac{\left|R_{target} - R_{actal}\right|}{R_{target}} \tag{10}$$

$$\text{BRA} = 1 - \text{BRE} \tag{11}$$

- Buffer Occupancy/Bit fluctuation and quality smoothness: Buffer occupancy analysis is also an essential part of a high-performance rate control algorithm. In the official reference software for HEVC and VVC, the buffers can be handled by enabling the HRD constraint [22,23] and the R-λ model. The buffer size is defined as

$$B_{uffer} = D_{elay} \times B_{width} \tag{12}$$

where $D_{elay}$ and $B_{width}$ represent the time delay and bandwidth, respectively. Buffer occupancy changes are generally represented graphically, and bit fluctuations and quality smoothness can usually be obtained in a graph comparing the bit values and PSNR values of different algorithms.

- Time complexity: Time complexity is a necessary metric for almost all algorithms. An optimized algorithm is bound to require low time overhead. The comparison time complexity is calculated by the following equation:

$$\Delta T = \frac{T_{pro} - T_{org}}{T_{org}} \tag{13}$$

where $T_{pro}$ represents the time consumed by the proposed algorithm and $T_{org}$ represents the time consumed by the reference algorithm.

## 3. Research Status of Rate Control

Before we begin to explore the latest developments in rate control, let's take a look at the general framework of the current study, as detailed in Figure 3. There are two main research trends: from method to process and from goal to process. The first one starts from the basic method and focuses on the performance of various video sequences in general, and the other one starts from the real environment and focuses on the performance of video sequences under current constraints.

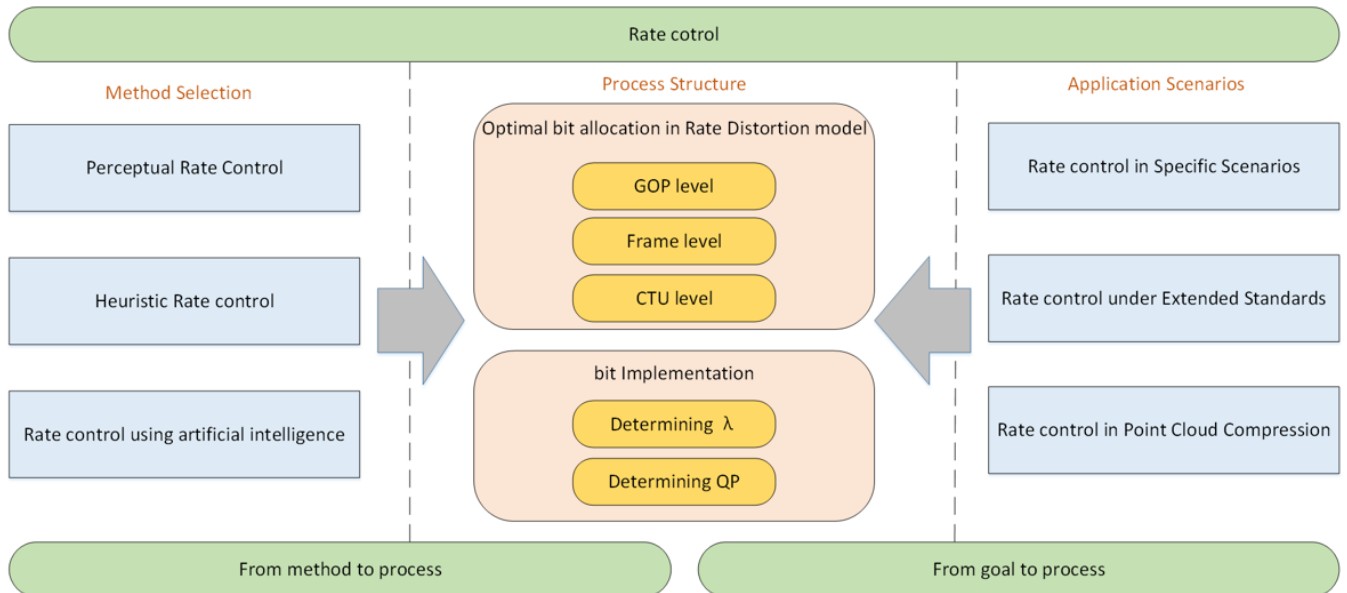

**Figure 3.** General overview of the rate control research.

Method selection: perceptual and heuristic methods are used, with the new artificial intelligence method emerging in recent years. Application scenario selection: The specific scenario and the extended standard scenario are more selected scenarios, and the point cloud is a new target scenario. This paper purposely takes the optimal bit allocation of the rate-distortion model as a subdivision field because the key part of rate control is bit implementation rather than bit allocation.

### 3.1. Optimal Bit Allocation for Rate-Distortion Models

An important part of the rate control algorithm based on the R-λ model is the bit allocation of three layers, namely, the GOP layer, frame layer, and CTU layer. The actual rate control algorithm can act on different coding unit levels, and the rate-distortion performance of different coding units is mainly related to the content characteristics of the coding units. Of course, there may also be interdependence in rate-distortion performance between different coding units, and many studies may propose related bit allocation schemes in two or even three layers. An important problem faced by bit allocation is the application under low-bit conditions, i.e., under the condition that there are not enough bits, how can we reasonably allocate as much as possible to ensure the quality of the reconstructed frame? Therefore, the study of optimal bit allocation has become an effective breakthrough point to improve coding performance.

In terms of GOP-level bit allocation, the LD configuration will have a greater impact on subsequent GOPs due to the large bit allocation occupied by the I frame, and the buffer may even be a negative value. The algorithm based on the R-λ model effectively solves this problem after the introduction of the sliding window, but there will still be some negative effects on the RA configuration, such as high cache area occupancy and large-period bit fluctuations. Song et al. [32] proposed to solve this problem by linearly reducing the buffer occupancy rate in each cycle to zero according to GOP; this method achieves



a slightly better rate-distortion performance without increasing the coding complexity. In the frame-level bit allocation, the frame-level and GOP-level bit allocation are closely related, so some scholars [33,34] consider the quality and hierarchical structure factors of the two models and use them for bit allocation. Guo et al. [35] proposed an R-D-dependent GOP-level bit allocation scheme, adopted a recursive Taylor expansion scheme to optimize frame-level bit allocation, and achieved good performance in LD configurations. The use of inter-frame prediction techniques leads to dependencies between frames, and the distortion of previous and subsequent frames will also affect the accuracy of bit allocation. Therefore, Li et al. [36] established a dependency model between frame distortion and bit allocation, extracted key features, and made the algorithm's peak signal-to-noise ratio increase and bit rate decrease by about 0.132 dB and 3.41%, respectively. In terms of CTU-level bit allocation, CTU-level bit allocation is often performed at the same time as frame-level. Additionally, Li et al. [37] proposed a content-based frame-level and CTU-level optimal bit allocation algorithm, which can better adjust the allocation bits of important pictures and corresponding blocks and improve the rate-distortion performance. Liu et al. [38] combined frame-level and CTU-level bit allocation as an overall optimization problem, obtained better bit rate allocation results, and improved coding efficiency. Block partitioning and intra-frame prediction techniques also create distortion dependencies between CTUs in a frame. Xie et al. [39] adopted a distortion propagation model to measure the temporal weights of CTU blocks, which effectively ensured the coding quality of these temporally correlated units, and Li et al. [40] proposed a recursive Taylor expansion method to measure the CTU level. Bit allocation improves the accuracy of rate control and is effective with videos with scene changes. Wang et al. [41] proposed a CTU-level bit allocation method for low-bit-rate applications based on regional classification which slightly improved subjective and objective quality.

On the whole, the bit allocation determines the initial optimization goal of each stage. Once a serious deviation from the goal estimation occurs, the stability of the entire sequence will be affected, so it needs to be carefully controlled.

### 3.2. Perceptual Rate Control

The quality of the video generally depends on the subjective guess of people, but since most videos are ultimately perceived by the human visual system (HVS), not all video distortions may be noticed. For this reason, many standards for evaluating video quality were born, such as MSE, PSNR (peak signal-to-noise ratio), SSIM (structural similarity), VMAF (video multi-method assessment fusion), and so on. At the same time, different video evaluation standards will have subjective and objective bias, as shown in Table 2.

**Table 2.** Comparison of common video quality measurement factors.

|  | Focus on Subjective Perception | Focus on Objective Perception |
|:---:|:---:|:---:|
| MSE |  | $\sqrt{}$ |
| PSNR |  | $\sqrt{}$ |
| SSIM/MS-SSIM | $\sqrt{}$ |  |
| JND | $\sqrt{}$ |  |
| ROI | $\sqrt{}$ |  |
| VMAF | $\sqrt{}$ | $\sqrt{}$ |

On the one hand, this series of standards is used as a reference index for the encoded video, and on the other hand, it can also be used as an adjustment parameter to establish a relationship model between distortion and bit rate. As a result, a rate control direction guided by the evaluation criteria emerges which largely pursues the overall objective quality of the video and tries to stabilize the subjective quality as much as possible.

Zhou et al. [42] proposed a split normalization scheme, tried to establish a connection between SSIM and coded bits, and applied it to the rate control of CTU. The result was a significant improvement in the performance of rate-aware distortion. Zhang et al. [43]

introduced a new visual analysis distortion, established a rate joint distortion model, and solved the rate joint distortion optimization problem with the Lagrange multiplier method; the experimental results show that the scheme is stable in use. The visual analysis performance can be improved by encoding in bits. Yuan et al. [44] proposed a hybrid video coding method based on distortion RDO and a rate control that mixes MSE and SSIM. This method divides the image into textured regular regions and textured irregular regions, with CTU as the minimum division unit, compared with HM16.14; the average BD-BR under AI, RA, and LD configurations is $-6.25\%$, $-7.53\%$, and $-9.05\%$, respectively, however, the disadvantage is that the complexity of the encoder is increased. Zhou et al. [45] also proposed a rate control method for HEVC based on just noticeable distortion (JND), which uses the JND factor as the weight of bit allocation to establish a rate-distortion model which improves the control accuracy and improves the coding performance. Additionally, Xiang et al. [46] proposed an adaptive Lagrange multiplier perceptual distortion model for intra-frame rate control to improve the overall quality of video compression.

In fact, in regard to the human eye, the video can choose to retain important area information as much as possible and reduce the proportion of unimportant information. For example, during a video call, improving the quality of the video about the face and reducing the quality of the edge environment can not only ensure a smooth call but also will not affect the user experience. This important area is called a region of interest (ROI). The focus of ROI-based rate control is to improve the overall subjective quality of the video. Meddeb et al. [47] proposed a rate control method based on the ROI which introduced regional bit allocation at both the frame layer and the CTU layer, and the scheme performed well for face perception. Maun et al. [48] integrated the ROI information into the existing rate control algorithm, and also considered the reference picture selection (RPS) method and the intra-frame refresh mechanism. As a result, the computational cost can be reduced while ensuring quality. Zhang et al. [49] proposed a rate control scheme in the ROI mode based on the DCT coefficient model which improved the PSNR value by an average of 0.5–1.0 dB compared to other rate control methods in ROI.

### 3.3. Heuristic Rate Control

As mentioned in the previous section, rate control is part of rate-distortion theory, and rate-distortion optimization itself is a convex optimization problem. In video coding, the Lagrange multiplier method is often used for extreme solutions under constraints, and this is only one of the tools to achieve the goal. The probability distribution is most closely related to the rate-distortion theory. Rate control process is based on the rate-distortion theory. It calculates the rate-distortion functions under different probability distributions, a series of empirical formulas and reference models are obtained, and then the DCT transform is used to process the luminance and chrominance coefficients to obtain a reliable simulation distribution. This process can explore available rate control model. In the past, widely used distributions included the Laplace distribution, generalized Gaussian distribution, and Cauchy distribution. Therefore, this paper classifies the rate control method combining theory and experiment as a heuristic rate control method.

Si et al. [50] proposed a CTU-level rate control method to adjust the quantization parameters based on the Laplacian distribution model of the transformed residuals, and the average performance gains under the LD and RA configurations were 5.0% and 2.4%, respectively. Hyun [51] proposed a VVC frame-level constant bit rate control method based on the recursive Bayesian estimation (RBE), which not only accurately estimates the rate but also assigns target bits based on the distortion variation of previously encoded frames with less fluctuation in visual quality. Chen et al. [52] proposed a quadratic rate-distortion model for frame-level rate control, and the proposed rate control algorithm can achieve 0.77% BD-BR reduction with similar control accuracy. Sanchez [53] used a piecewise linear approximation to approximate the slope of the R-D curve. In the AI and AI-SCC configurations, it achieved higher accuracy and a more constant number of bits per frame than HEVC's own algorithm. Mao et al. [54] proposed a VVC rate control method based

on the composite Cauchy distribution. Based on the derived R-Q model and D-Q model, the relationship between the rate, distortion, and coding parameters was established. An adaptive bit allocation method was then proposed which can achieve 1.03% BD-rate saving in LDB configuration and 1.29% BD-rate saving in RA configuration.

*3.4. Rate Control Using Artificial Intelligence*

The victory of AlphaGo, the success of unmanned driving, the breakthrough of pattern recognition, and the rapid development of artificial intelligence have amazed countless people. As the core of artificial intelligence, machine learning has also attracted much attention in the development of artificial intelligence. Today, the application of machine learning has spread to various branches of artificial intelligence, such as expert systems, automatic reasoning, natural language understanding, pattern recognition, computer vision, intelligent robots, and other fields. In recent years [55], the popularity of machine learning in the field of video coding has also increased. The use of various machine learning tools to solve optimization problems is also a major research direction. Especially in rate control, machine learning can discover rules from huge data sets, and, by extracting various features, a fitted model can be trained to replace the rate-distortion model, resulting in more accurate bit estimation and better rate-distortion performance.

3.4.1. Traditional Machine Learning Methods

Traditional machine learning methods can more accurately predict bit allocation and related parameters with the help of classical models or algorithms. Wang et al. [56] used four traditional machine learning methods: support vector regression (SVR), random forest regression (RFR), gaussian process regression (GPR), and artificial neural networks (ANN) to extract four highly descriptive features to capture the relationship between the video content and rate-distortion model and improve the accuracy by 8.65% with an affordable computational complexity increase. Gao et al. [57] proposed an optimal interframe mode rate control method based on support vector machine (SVM) to classify and assign bits to CTU of different frames, which mainly solved the problem of unreasonable bit allocation. While maintaining the stability of bit rate, the average PSNR value also increased by 0.18 dB. Hsieh [58] et al. proposed a machine learning method with low hardware complexity, and the average PSNR increased by 0.82 dB; the average BD-BR increased by 15.62%. Wei Gao [59] et al. used the RBF kernel-based $\varepsilon$-SVR method for initial QP prediction, and achieved significant progress in rate-distortion performance, reflecting the effectiveness of machine learning methods. In addition, Sun et al. [60] proposed a second-order model to simulate the relationship between bit rate and constant speed factor under the framework of quadratic coding, extracted machine learning features, and used it to accurately estimate model parameters.

3.4.2. Deep Learning Methods

The convolutional neural network (CNN) has become a popular neural network in recent years. It has achieved great success in the fields of artificial intelligence and signal processing, and also provided a novel and promising solution for image and video compression. The network is well-trained based on massive image and video samples labeled for specific tasks in an end-to-end strategy. The trained neural network can then solve the classification, recognition, and prediction tasks of test data well, and has efficient adaptation. In recent years, CNN-based deep learning methods have also been gradually applied in rate control. The deep learning method can extract the pixel-related information of each frame or each CTU and input it into the CNN network to realize bit allocation and bit control. Lu et al. [61] proposed a rate control method for HEVC intra-frame coding based on the CNN, applying a CNN at the CTU level to achieve accurate QP prediction while maintaining the same reconstructed image quality, reducing the BDBR by an average of 1.33%. They also proposed a CTU-level bit allocation and bit control method based on a CNN. The training of the CNN was done using a natural image UCID dataset and



RAISE dataset and predicted the model parameters of each CTU. The BD-rate of the Y component was reduced by 0.7% on average, and the BD-rate of the U and V components was reduced by more than 2%. Xu et al. [62,63] designed a rate-distortion modeling method for HEVC that differs from the traditional CNN with a full CNN. By learning an end-to-end pixel-to-pixel map from the original image to the SSIM map, representing the distortion, it proved the possibility of achieving an R-D relationship. Cheng et al. [64] proposed a rate control method called the content adaptive rate factor (CARF) at the GOP level on the x265 encoder to adjust the constant rate factor (CRF) value of each GOP. Compared with the average bit rate (ABR) mode of x265, CARF's method reduces PSNR, VMAF, and SSIM by 4.12%, 5.35%, and 5.73%, respectively.

### 3.4.3. Reinforcement Learning Methods

Reinforcement learning is a branch of machine learning used to describe and solve problems in which agents learn strategies to maximize rewards or achieve specific goals in the process of interacting with the environment. The fact that reinforcement learning has been around for decades shows that it is not a new technique. Deep reinforcement learning is particularly suitable for high-dimensional state spaces but existing reinforcement learning methods are very difficult with regard to the design of feature selection. However, since deep reinforcement learning can learn the main features of the data from different levels, it can successfully solve complex tasks only with underlying prior knowledge. Common deep learning algorithms include the DQN (deep Q network), A3C (asynchronous advantage actor-critic), DDPG (deep deterministic policy gradient), SAC (soft actor-critic), TD3 (twin delayed DDPG), PPO (proximal policy optimization), etc., and can solve a large number of discrete and continuous problems. When reinforcement learning is combined with rate control, a new structure is created, as shown in Figure 4. It is mainly composed of five parts (agent, environment, action, reward, and state). The agent selects an action for the environment, and the environment accepts the action and changes its state while at the same time generating a reinforcement signal (reward or punishment) back to the intelligence, which then selects the next action based on the reinforcement signal and the current state of the environment; the principle of selection is to increase the probability of receiving positive reinforcement (reward). Rate control is viewed as a Markov decision process (MDP) problem, which can be solved by the Bellman equation.

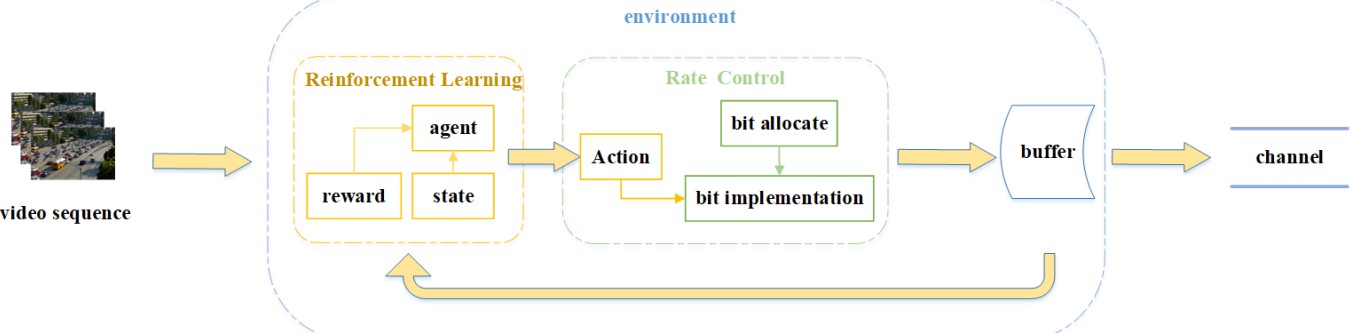

**Figure 4.** RL-based rate control framework.

The reinforcement learning model will replace the rate-distortion model for reasonable allocation. With the previous rate control process, we find that the QP decision is a discrete problem. The reinforcement learning method suitable for solving discrete problems is used to train the model. According to the state information of each frame or each CTU before coding, the corresponding action (QP) is made, and good performance is finally achieved. However, the exact set up of the environment and the choice of reward are also difficult to achieve to make reinforcement learning algorithms effective. Hu et al. [65] first proposed applying the reinforcement learning method of Q-learning to HEVC rate

control. The basic task was to determine the quantization parameters at the CTU level. As an environment, to train a neural network model, the performance of this method is comparable to that of a well-optimized representation algorithm, initially showing the potential of reinforcement learning in the field of rate control. Zhou et al. [66] introduced the A3C algorithm for dynamic video sequences and applied it to frame-level and CTU-level rate control, mainly through offline learning and prediction methods to extract action unit features from the encoder. Controlled performance indicators were used to assign action weights and finally determine the frame-level and CTU-level quantization parameters, achieving a better subjective and objective quality and smaller bit rate errors than the reference software HM16.19. Ho et al. [67] implemented the DDPG method in the x265 encoder and established a dual-criticism network framework for two types of criticism: rate criticism and distortion criticism. This method can achieve a 4.9–5.0% bit rate savings and BD-PSNR gain of 0.22–0.24 dB. Ren et al. [68] developed methods based on deep reinforcement learning combining the frame and region of interest (ROI) bit allocation processes and proposed an ROI-based DDPG rate control algorithm. This method improved the quality of ROI with stable bit rate fluctuations when VVC encoding was performed for game videos. Gao et al. [69] also proposed a PPO algorithm to train a deep reinforcement learning model and used it for the low-latency video communication aspect of HEVC. This method is not only used for QP decisions but also applied to target bit allocation, which achieves better low-latency transmission performance while maintaining the accuracy of rate control.

### 3.5. Rate Control in Specific Scenarios

Different application scenarios will have different requirements for rate control, so the optimization conditions and goals are also different. For specific scenarios, rate control still needs to design new algorithms to adapt. Li et al. [70] proposed a regional-level and CTU-level optimized bit allocation method in the scenario of logo insertion into video, which saved an average of 4.46% of BD-rate, while the speed loss was only 2.26%. Li et al. [71] proposed a weighted CTU-level bit allocation algorithm for a projected 360-degree video compression scene, considering the matrix projection format, and established a more stable rate control model. The relevant 360-degree video index results showed that the method has a better subjective and objective quality and less bit error. Victor Sanchez et al. [72] specially performed an ROI lossless coding rate control for digital pathological images which could accurately obtain the overall bit rate. Zupancic et al. [73] proposed a two-pass rate control method for the scenario of ultra-high-definition television (UHDTV). In comparison with the variable bit rate mode of the customized HM fast codec, an average lower compression loss was achieved.

High dynamic range (HDR) video can provide a more dynamic range and image details than ordinary images and can better reflect the visual effects of the real environment. HDR video usually uses 10-bit encoding and is currently being researched and selected more. Bai et al. [74] established a CTU-based brightness-bit relationship model for HDR video which can achieve a $-4.4\%$ BD-rate improvement. Daniel et al. [75] proposed a rate control method of multiple R-$\lambda$ models, using a set of three different R-$\lambda$ models to correspond to three types of rate-distortion feature regions distinguished by brightness, especially for areas with high brightness, to increase the reconstruction quality of HDR content. Mir et al. [76] also proposed a new $\lambda$-QP relationship model suitable for HDR content which can well estimate the relationship between HDR distortion and the bit rate used. For HDR video test sequences, the method achieved up to 1.36 dB average PU-PSNR improvement. Zhou et al. [77] proposed a rate control algorithm based on the rate-distortion model of the visual difference predictor, which was used for parameter estimation and reducing the rate error. The results of multiple HDR indicators can reduce the bit rate by about 3% on average.

### 3.6. Rate Control under Extended Standards

As early as H.264/AVC, there have been scalable video coding (SVC) and multi-view video coding (MVC) to provide an extended version of HEVC and also establish the Joint Collaborative Group for the Development of 3D Video Coding Extensions (JCT-3V) to work on multi-view and 3D video coding extensions to other video coding standards for HEVC. The second version of HEVC was completed in October 2014. In the second version, JCT-VC proposed a format range extension (RExt) and scalability extension (SHVC) successively, while JCT-3V proposed a multi-view extension (MV-HEVC). In the third version, completed in February 2015, JCT-3V proposed a 3D-HEVC. The latest VVC standard has included extended content. These extended standards mainly deal with the emergence of new video formats such as UHD, high dynamic range, and wide color gamut. Li et al. [78] proposed the λ-domain bit rate control algorithm of SHVC which provides an optimal bit allocation method for each layer and has a smaller rate error and better rate-distortion performance. Fiengo et al. [79] treated frame-level bit allocation as a convex optimization problem and proposed an efficient algorithm to achieve MV-HEVC rate control, which has better performance in rate-distortion. Li et al. [80] proposed a new CTU-level R-λ model parameter prediction method, derived an accurate power model to represent the target bit rate relationship between the base view and the slave view, and developed a new linear model to allocate the target bit rate of P-frames in the slave view; the overall method outperformed other advanced methods. Abolfathi et al. [81] proposed a new method for a free-view video rate allocation based on MV-HEVC. The appropriate bit rate was allocated to each view through the distance between different view directions. This method is similar to the rate control method in the λ domain. Compared with the algorithm, it can achieve higher coding efficiency. Tan et al. [82] proposed a new dependent view distortion model to study the dependence relationship between the synthetic view and the encoded view, make relevant bit allocation and an effective initial QP decision, and finally achieve a better rate-distortion performance than the standard algorithm of 3D-HEVC. Song et al. [83] improved the 3D-HEVC algorithm according to the prediction weight generated by the MAD prediction error and proposed a CTU-level rate control algorithm based on the weight-based R-λ model, which is similar to the rate control benchmark based on the R-λ model in the HTM software. Compared with the algorithm, it has higher rate control accuracy and rate-distortion performance.

## 4. Discussion

Rate control techniques have been around for a long time, but the overall mode of operation has not changed significantly. As shown in Figure 5, even though the reference software has transitioned from HM to VTM, the reference software in VVC, like HEVC, still uses the R-λ model. This also proves that the R-λ model performs better compared to other models. Therefore, most studies have been based on this model for algorithm improvement. When the R-λ model is truly abandoned, the model-free approach may become mainstream.

To demonstrate the effectiveness of rate control, we tested video sequences with different resolutions using the reference software JM [84], HM [85], and VTM [86]; the corresponding versions are JM19.0, HM16.26, and VTM18.0, respectively. The YUV video sequences used for the experiments were all taken from the official recommended videos [28]. The experiments set the target bit rate to 100,000, the number of input video frames to 50, and the encoding configuration to include LDB, LDP, and RA, as detailed in Figure 6. Only the JM software uses the default configuration and does not support higher resolution video. Among the SDR video sequences chosen are RaceHorses ($416 \times 240$), PartyScene ($830 \times 480$), ChinaSpeed ($1024 \times 768$), vidyo1 ($1280 \times 720$), Kimono1 ($1920 \times 1080$), and Traffic ($2560 \times 1600$). From Figure 6a, it can be seen that the average PSNR values of VTM and HM are much higher than those of JM. The PSNR values of both LDB and LDP do not differ much from each other. From Figure 6b, the BRE values of VTM and HM are generally smaller than the BRE values of JM, and the degree of BRE variation is greater under the

RA configuration. The average BRE of the reference software is now around 0.001, which means that the bit allocation accuracy is already very high. The actual time spent shows VTM > HM > JM. It is worth mentioning that the time complexity increase has little to do with the rate control process. In general, the performance of the reference software HM and VTM are significantly better than that of JM in all aspects when the rate control is on, which is also the result of the difference between the models. The reference software HM and VTM both use the R-λ model with little difference between them (see Appendix A for specific test effects). (If you carefully observe the background, you will find that the image quality from left to right improves).

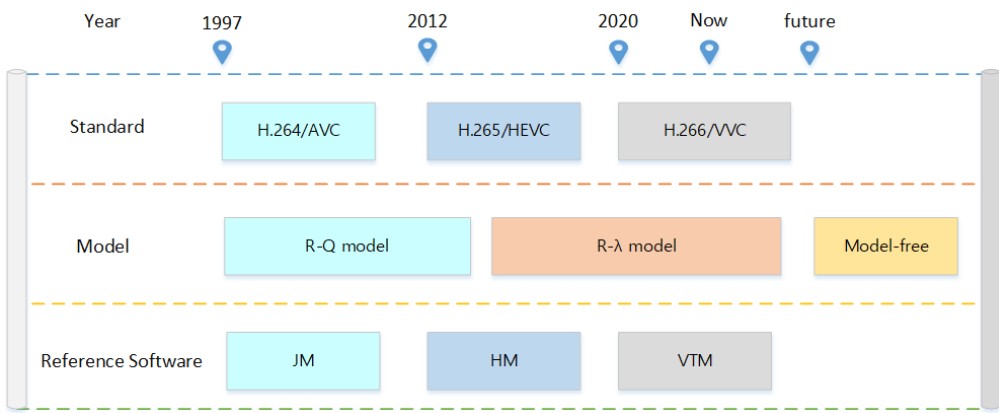

**Figure 5.** Model changes in official reference software for SDR video.

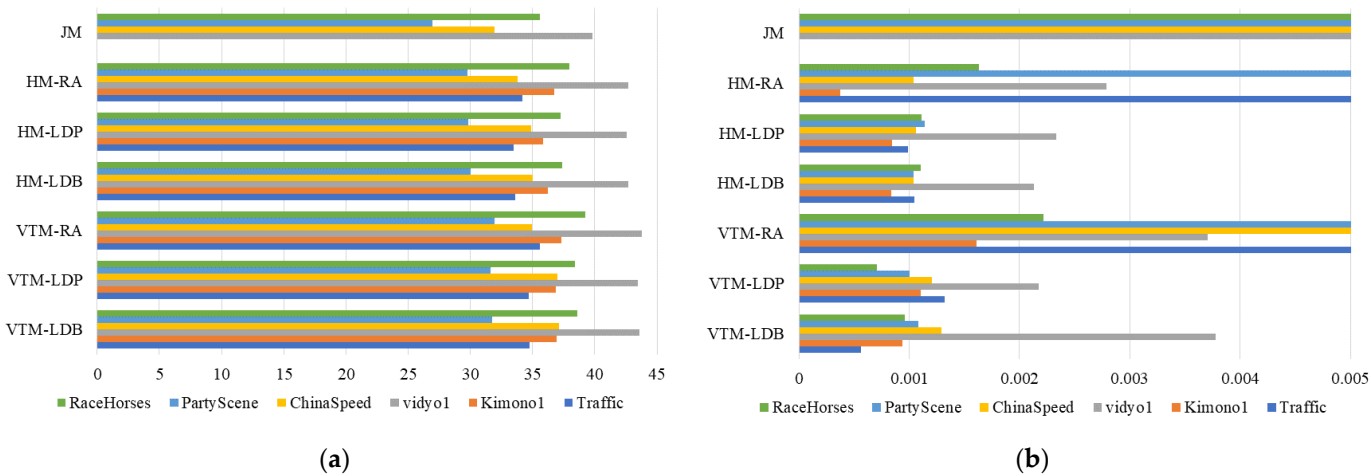

**Figure 6.** (**a**) PSNR values for each coding configuration; (**b**) BRE values for each code configuration.

The goal of rate control experiments in video coding can be different from other experiments, for experiments such as intra-frame prediction or inter-frame prediction, the goal is to save time in the rate-distortion process with controlled losses. However, the goal of rate control is to improve the rate-distortion performance under various constraints. For those videos with large scene changes, the enhanced effect of bitrate control can only be fully realized.

As shown in Tables 3 and 4, the current research on rate control has gradually transformed from HEVC to VVC. This is because the use of HM reference software as an experimental reference is still dominant. Now, the research on the reference software of VVC is still in its infancy. Almost all studies will use the hierarchical structure to carry out comparative experiments and various algorithms to pursue an average BD-rate with higher negative values and lower RBE values. Their algorithms will almost always compare the official reference software HM and VTM, and among the three known levels of bit allocation,

the frame level and CTU level are the most studied, and the GOP level is relatively less. A comparison of standard reference software experiments reveals that some experiments can achieve significant performance improvements within an acceptable range of reduced bit rate accuracy. Similarly, the bit allocation accuracy in RA configurations will vary greatly.

**Table 3.** Performance comparison of RC algorithms in recent years (1).

| Method | Anchor | Configuration | Level | Hierarchical Structure | Average BD-Rate (%) | BRE (%) |
|---|---|---|---|---|---|---|
| Guo 2019 [35] | HM16.17_RC | LDB/LDP | Frame | Enable/disable | −3.70/−3.20 | 0.061/0.072 |
| Li 2018 [36] | HM12.0_RC | RA | Frame | Enable | −3.41 | 6.80 |
| Zhang 2019 [33] | Li 2018 [36] | LDB/LDP | GOP | Enable | −5.80/−5.07 | 29.87/28.41 |
| Zhou 2019 [42] | HM16.19_RC | LDB | CTU | Enable/disable | −3.10/−5.0 | 2.86/0.69 |
| Zhou 2020 [45] | HM16.19_RC | LDB | Frame/CTU | Enable | −3.30 | 2.30 |
| Mao 2021 [54] | VTM3.0_RC | LDB/RA | GOP/Frame | Enable | −2.96/4.36 | 0.35/2.18 |

**Table 4.** Performance comparison of RC algorithms in recent years (2).

| Method | Anchor | Configuration | Level | Hierarchical Structure | Average BD-Rate (%) Y/U/V | BRE (%) |
|---|---|---|---|---|---|---|
| Li 2017 [62] | HM16.9 | AI | CTU | / | 1.10/4.30/4.50 | 1.07 |
| Hu 2018 [65] | HM16.15 | AI | Frame | / | 2.80/−0.30/0.30 | 0.27 |
| Chen 2020 [52] | VTM2.0_RC | AI | Frame | / | −0.77/-/- | 0.06 |
| Zhou 2020 [66] | HM16.19 | LDB | Frame/CTU | Enable | −3.60/−0.90/−0.10 | 0.05 |
| Liu 2021 [38] | Li 2018 [36] | RA | Frame/CTU | Enable | −0.91/−1.50/−1.23 | 0.17 |

## 5. Challenges and Prospects

### 5.1. End-to-End Rate Control

Because the traditional R-λ model follows a fixed bit allocation and bit implementation strategy, it always has limitations in the performance of videos with large complexity differences. Gao et al. [87]. then found that there can be an imbalance between intra- and inter-frame coding. This is because it is difficult to capture the overall quality when encoding. Zhou et al. [88] proposed a Lagrange multiplier approach to obtain optimal closed-form solutions of quantization parameters of coding units starting from the perspective of solving convex optimization problems and achieved good results. If a model-free method can be used instead of a model-based method, and a series of decisions can be directly completed by directly extracting image information, it may become a major breakthrough in the future. At present, artificial intelligence algorithms can be used as an alternative method. However, there are still many difficulties to achieve this goal.

### 5.2. Rate Control for Various Types of Videos

Most studies use standard test sequences, exemplified by VVC. The standard test sequence is divided into A1, A2, B, C, D, E, and F categories. These videos are mainly composed of 480 p, 720 p, 1080 p, 2 k, and 4 k. Therefore, for ordinary videos, this paper only considers the size of the resolution and the frame rate. In fact, there are still many kinds of videos worthy of in-depth studies, such as HDR video, projection 360-degree video, 3D video, VR video, and so on. There has been a lot of related research on HEVC in the past, but it is a difficult problem to extend the VVC rate control to these fields. Zhao et al. [89] achieved bit rate savings of up to 11.77% in the latest standard VVC for 360-degree video types divided into different regions at the frame level according to demand. Although the exploration has not really begun, multiple types of videos relevant to rate control have the ability to be applied to VVC.

### 5.3. Perceptual Rate Control Method Based on VMAF

Among them, VMAF, as a new evaluation index, adopts the SVM-based nuSVR algorithm. During the running process, according to the pre-trained model, each video feature is given different weights, and a score is generated for each frame. Finally, the average algorithm is used to summarize, and the final score of the video is calculated, which ranges from 0 to 100. At present, VMAF is widely used in rate-distortion optimization [90–94], but it is rarely used as a guide for rate control; most of them are used to evaluate performance. VMAF is a new and mainstream objective video evaluation standard. In terms of perceptual rate-distortion performance, it has advantages over single PSNR and SSIM, so it shines in rate-distortion optimization. Rate control is in turn closely linked to rate-distortion optimization. Therefore, we think it is a good choice for use in the perceptual rate control algorithm.

### 5.4. Rate Control Method Based on Point Cloud

In the auto drive system, lidar is an environmental sensing device. The point cloud data collected by lidar plays an important role in detecting three-dimensional targets, feeding back whether there are nearby obstacles, and relaying how far away objects are from the front of the vehicle. The research on point cloud compression is also the latest focus. Li et al. [95] proposed a point cloud compression algorithm based on geometry and achieved good performance in G-PCC reference software. Liu et al. [96] proposed a region-based 3D point cloud compression algorithm with an average bit rate error of only 3.7%. Wang et al. [97] predicted basic unit (BU) parameters through a CNN-LSTM neural network to improve rate-distortion performance and subjective dynamic point cloud quality. Rate control algorithms related to point cloud compression are also gradually attracting attention.

### 5.5. Application-Oriented Rate Control

As we all know, there is a large gap between reference software and industrial software, because (1) the audiences are different and therefore have different needs, and (2) reference software is closer to the theoretical level than industrial software and often cannot take into account the practical difficulties encountered. The process of establishing a future fast VVC encoder is also still advancing [98]. For real-time mobile video coding, Hsieh et al. [99] implemented a rate control design related to motion estimation based on an improved machine learning scheme. Hsieh et al. [100] also proposed a hardware-oriented CBR control algorithm which was used for H.265/efficient video coding of a multiprocessor system on chip (MPSoC). Quality of experience (QoE) [101] is often considered a goal for real-time video streaming applications. Therefore, most of the current video coding rate control problems are source coding optimization control problems under the assumption of the given channel bandwidth, and the future research direction also includes the joint optimization of the channel and source coding in many heterogeneous network environments. Application-oriented bit rate control will also be a focus in the future.

## 6. Conclusions

As an important module in the whole process of video coding, rate control is used to find a balance by coordinating the relationship between rate and distortion on a macro level, so as to solve the problems of excessive video quality fluctuations in various complex scenarios. Rate control is mainly composed of two steps, bit allocation and bit realization. Bit allocation includes GOP-level bit allocation, frame-level bit allocation, and CTU-level bit allocation. Bit implementation is mainly accomplished by adjusting various parameters to determine the frame-level and CTU-level quantization parameters, and then handing it over to the encoding process to complete. In recent years, the rate control algorithms of HEVC and VVC have not changed much on the whole and are still designed with the R-$\lambda$ model as the main body. Although VVC has a great improvement in technology compared to HEVC, due to its own coding structure, the complexity of the VVC encoder is 7.5 times

or even 34 times that of the HM encoder in different configurations [102], which means that the time cost will be more expensive to study VVC code in the future and is the biggest obstacle to rate control technology. At present, the exploration of the rate control algorithms of HEVC and VVC is multi-directional, with some focus on SDR video [28] and some focus on HDR video [103], while some focus on the subjective perceptual rate control and others focus on the objective rate control. In addition, some focus on basic paper methods, while others focus on new methods of artificial intelligence. From the perspective of the entire research status, the current VVC rate control research is still in its infancy, and there are few pieces of literature available for reference. Therefore, it would be a good choice to migrate the mature theory of HEVC to VVC.

**Author Contributions:** Funding acquisition, H.Z.; Resources, H.Z. and J.X.; Supervision, H.Z.; Writing-original draft, H.Z., J.X. and S.H.; Writing—review and editing, H.Z., S.H., C.S. and Z.D. visualization, H.Z., J.X. and S.H. All authors have read and agreed to the published version of the manuscript.

**Funding:** This work was supported by the Hainan Province Key R&D Program Project (No. ZDYF202 1GXJS010, ZDYF2019010), the National Natural Science Foundation of China (No.61562023, 61362016, 61502127), the Major Science and Technology Project of Haikou City (No.2020006), and the Hainan Provincial Natural Science Foundation of China (720RC616).

**Institutional Review Board Statement:** Not applicable.

**Informed Consent Statement:** Not applicable.

**Data Availability Statement:** Not applicable.

**Conflicts of Interest:** The authors declare no conflict of interest.

**Appendix A**

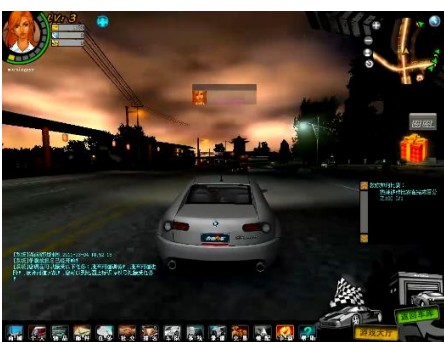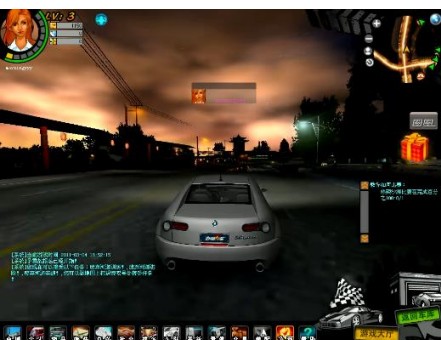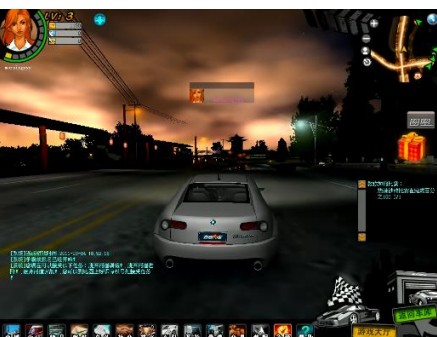

**Figure A1.** The images are all derived from the seventh frame of the sequence ChinaSpeed (from **left** to **right** are the results after coding by JM, HM, and VTM reference software).

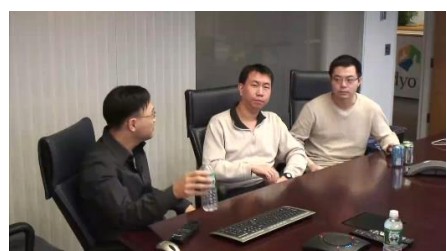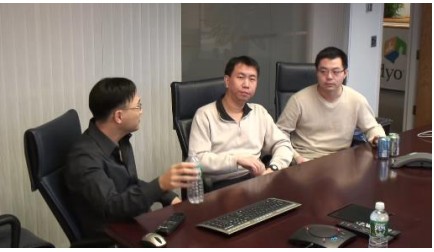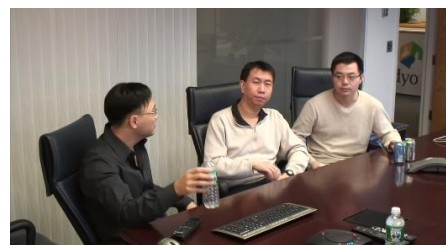

**Figure A2.** The images are all derived from the seventh frame of the sequence vidyo1 (from **left** to **right** are the results after coding by JM, HM, and VTM reference software).

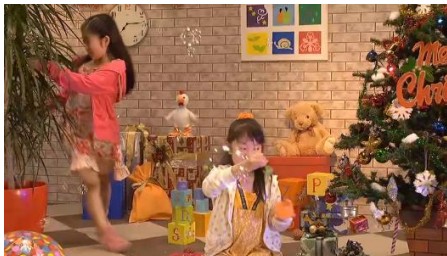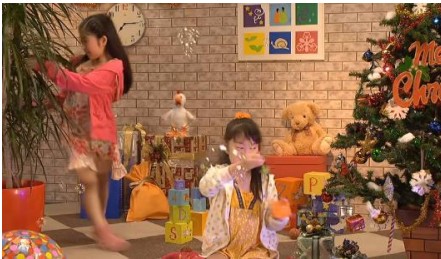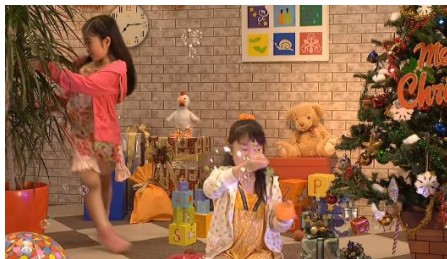

**Figure A3.** The images are all derived from the seventh frame of the sequence PartyScene (from **left** to **right** are the results after coding by JM, HM, and VTM reference software).

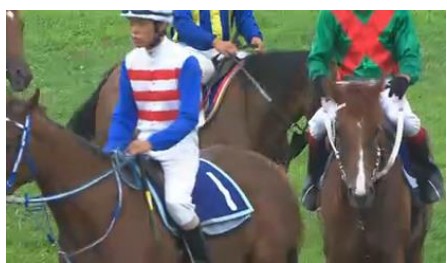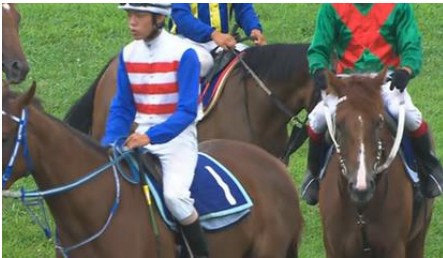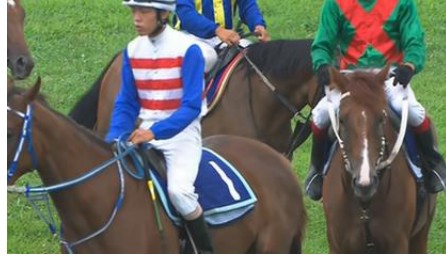

**Figure A4.** The images are all derived from the seventh frame of the sequence RaceHorses (from **left** to **right** are the results after coding by JM, HM, and VTM reference software).

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
