# Peer review of "Rate Control Technology for Next Generation Video Coding Overview and Future Perspective"

_electronics, doi:10.3390/electronics11234052_

Round 1

Reviewer 1 Report

The paper is about the next generation of rate control in Video Coding. The authors summarize the rate control models implemented in different video coding standards highlighting their complexity and quality. 

-The discussion part is too short and doesn't discuss the outcomes of each implementation; moreover, the authors didn't run an experiment to highlight the performance of one rate-control model over the others.

- The authors summarized the problems and difficulties of different rate control models and suggested a few approaches to enhance and solve these difficulties.

It would be great if the authors designed an experiment to measure the performance of each model, unifying the common factors and highlighting a proper estimate of different factors

Author Response

Hello, thank you for pointing out the shortcomings of the article. I also realized that it was unreasonable for the discussion section to be a separate chapter without substantial content. To this end we have added more content in this section. The changes to the Discussion section occur in Chapter 4, as detailed in lines 540-592. A separate chapter and additional citations have also been added to the Future Outlook section, as detailed in Chapter 5, lines 594-652 of the article.

I also notice that you want to be able to compare the performance of different rate control models. In fact, the choice of official reference software has given the answer, R-λ model is HEVC and VVC consistent use of the model, so the performance is completely superior to R-Q model and R-ρ model, now most research is also around this model, the future may be different. For this explanation, I also added the development diagram of the model. For the experimental part you wish to add, I added a comparison of the performance of the reference software rate control in AVC, HEVC, and VVC. At the same time, compared with the previous table, we also expand more new discussion. In addition, the outlook for the future is also replanned and provided with detailed reference basis, I hope to get your approval.

The above is our answer to specific questions, please criticize and correct where it is not in place.

Reviewer 2 Report

Title

Modified the title to reflect high efficiency video coding or versatile video coding and systematic literature review  

Motivation

The lacks strong motivation for conducting the LR, please provide a strong motivation situated within literature for conducting the LR.

Methodology

The authors claim that it is a comprehensive literature review, thus, they should add review methodology. Refer to the following papers for guide on how to develop the methodology

Jauro, F., Chiroma, H., Gital, A. Y., Almutairi, M., Shafi’i, M. A., & Abawajy, J. H. (2020). Deep learning architectures in emerging cloud computing architectures: Recent development, challenges and next research trend. Applied Soft Computing96, 106582.

Algorithm basic concept

The theories with mathematical background of the main algorithms should be provided in the paper to make it self-contained and show the readers how the algorithms operate to achieve it goal  

Taxonomy

Create a taxonomy of the papers based the algorithms, theories, applications, findings or any other perspective from the view point of the researchers and discuss it

Discussion

The discussion is too shallow and lacks sufficient takeaways. The authors should please write a comprehensive discussion with sufficient takeaways. The table in the discussion section is not necessary. In the discussion section provide enough analysis with visual representations such as figures, tables, etc.

Case study

I couldn’t find case studies in the paper, provide at least 4 case studies where the technology is currently in use in the real world. As such, create new section for the case studies and discuss them.

Challenges and future research

Create a new section before the conclusion and move the challenges and future research direction in the conclusion to the new section challenges and future research. Then, add more challenges and future research directions because the  research challenges currently in the paper is too small for a comprehensive review

quality assessment

Assess the quality of the LR conducted to give confidence to the readers that the RL passed the quality assessment criteria. Refer to the following paper to see the guide for assessing LR: Kitchenham, B., Brereton, O. P., Budgen, D., Turner, M., Bailey, J., & Linkman, S. (2009). Systematic literature reviews in software engineering–a systematic literature review. Information and software technology51(1), 7-15.

References

Add more 2022 papers

Author Response

Hello, I have read your reply completely, and thank you for your specific classification of the corresponding suggestions, so I will also reply to you later. For the two articles you mentioned earlier, I also compared their organizational structure and discussion methods in detail. I hope that I can satisfy you with the changes to the articles.

The title. Following your suggestion, we have revised the title.

The motivation. According to your requirements, I have revised the title, structure, discussion content and so on, and tried my best to make this article meet the LR standard. For similar literature review, please refer to the following LR: Hussain, A.J.; Ahmed, Z. A survey on video compression fast block matching algorithms, Neurocomputing, 335, 215-237

Basic concept of algorithm. In my opinion, the core concept that rate control can grasp is rate distortion theory. Therefore, based on this theory derived R-Q model, R-λ model algorithm is the current countermeasures, because the latest standard reference software is used in R-λ model, so I provide its algorithm flow in detail.

Discuss. The changes to the Discussion section occur in chapters 4, as detailed in lines 540-592. In the discussion section we re-added the development of the current rate control model. I also tested the rate control with various official reference software. This is also the target algorithm of the study comparison. We also combined it with the previous table for comprehensive analysis and gave more results analysis and discussion.

Case studies. In the case study, I listed typical examples such as deep learning and reinforcement learning, and also prospected the practical application in terms of the research direction.

Classification. In my opinion, the current research status is difficult to be classified from a specific aspect, which is the reason why few literature reviews in this field have been done so far. Even if each literature uses the same kind of algorithm, it will produce quite different results due to different video objects, different bit allocation levels, different experimental conditions, etc.

Challenges and future research. Chapter six is listed separately. The changes are made on lines 593-652.Considering your opinion, I set up a separate chapter to describe it. For this purpose, I also add more citing arguments and new research direction. This new research direction is also at the application level, after all, there is still a long distance between theoretical reference software and mature industrial software.

Quality assessment. I also read the quality assessment in your paper. You mentioned a literature review at the software level. My answers to the four questions in this part are Y, P, Y, P.

References. I took your comments into consideration and added more 2022 literature and cited it where appropriate. The added references are [46], [54], [87], [88], [89], [97], [100], which are located in lines 793, 814-815, 898-899, 900-901, 902-903, 923-924, and 929-930.

The above is our answer to specific questions, please criticize and correct where it is not in place.

Reviewer 3 Report

comments to Authors 

Major revision:

In 3.4. Rate control using artificial intelligence, no citation why?

Discussion chapter was too short.

Include the relative discussions in a table between the proposed technology or processes. 

Include year-wise discussion with suitable plots. 

Summarize the video transmitting technologies with low band width networks.

Future scopes should be included with suitable citations. Reasons ..

Plot the year-wise developments of each method. 

Author Response

Hello, thank you for reading my article in detail and giving me valuable suggestions. The major changes are in chapters 4 and 6, lines 540-592 and 594-652. I summarized my answer in the following points:

  1. In section 3.4, thanks for pointing out the problem, I did forget to add the reference evidence, now I have added it again. It happens on line 376 of the article.
  2. In the discussion part, I also redesigned a group of experiments on models to show the differences in effects between models. Detailed information about the reference software in the study is added. The effects of different configurations are also discussed. This change occurs in lines 549-567 of the article. Of course, I also combined the results of the research status paper for analysis and discussion. It's in lines 581-592. Technology or process is not included in the discussion because these articles have already been analyzed in the previous status quo. Then for the annual discussion, I have a detailed diagram of the current rate control model. It's at the beginning of the discussion, lines 540-546 of the text. I think model-based discussion can make the table content more acceptable, and standard experiments can also add credibility.
  3. Low bandwidth network video transmission technology. Low bandwidth video transmission technology actually belongs to the category of channel coding, and my current research belongs to the category of source coding. Therefore, I am sorry that the low bandwidth video transmission is not within the scope of my discussion. But I will provide you with a case of low bandwidth transmission in this field. Rahmati, Z. Qi and D. Pompili, "Underwater Adaptive Video Transmissions using MIMO-based Software-Defined Acoustic Modems," in IEEE Transactions on Multimedia.
  4. I will devote a separate chapter to the future outlook. Moreover, the latest literature is also added to demonstrate and provide detailed reasons. The specific changes are made on lines 593 - 652.
  5. Annual development of each method. The fact is that other approaches besides artificial intelligence have been around for a long time, and mapping out the path forward in detail is not easy. But the discussion of models I think is convenient and appropriate. Because almost all methods rely on rate-distortion models. It's an indispensable part.

The above is our answer to specific questions, please criticize and correct where it is not in place.

Round 2

Reviewer 2 Report

The paper has been improved. However, major work still required. The case study and quality assessment were not properly responded. 

Author Response

Hello, thank you for your response to the previous revision. I'm sorry if my earlier answer was not appropriate or detailed enough. I'll try to be more accurate this time.

Case studies:In real life, this technology is involved in online video, live streaming, real-time video conference, etc. I think these are the case studies you need. But my research in this paper is mainly based on the basic applications of the official reference software JM, HM, VTM, which are AVC, HEVC, VVC (the latest) and other video coding standards. At present, only industrial encoders x.264 (AVC) and x.265 (HEVC) are used in the real world. This means that there is still a long way to go from current research to practical application, and I can only list it as a research method at present. At the same time, I added Appendix A, which will show the effect of coding with different official reference software.

Quality assessment: I think the current overview is in line with the overview of the field of video coding. Just like the relationship between modules and software, rate control is only a small part of video coding. The research I have done is all around this aspect. At present, few scholars have made a complete review on this aspect, so I want to solve this pain point. The latest relevant articles are as follows: Guo H.W., Zhu C., Z Y.M. Research progress of HEVC rate control technology [J]. Journal of Chongqing University of Posts and Telecommunications (Natural Science Edition), 2018, 30 (02): 199-209. In fact, it is out of date, and I think the work I do is more in line with the current needs.

Reviewer 3 Report

Accepted present form..

Author Response

Thank you very much for accepting my form. I really appreciate your comments on the article before.